



# 1 Emissions of greenhouse gases from energy use in agriculture,

# 2 forestry and fisheries: 1970-2019

Alessandro Flammini[1], Xueyao Pan[2], Francesco N. Tubiello[2]*, Sally Yue Qiu[3], Leonardo
Rocha Souza[4], Roberta Quadrelli[5], Stefania Bracco[6], Philippe Benoit[3] and Ralph Sims[7]
[1] United Nations Industrial Development Organization, Department of Environment, Vienna, Austria
[2] Food and Agriculture Organization, Statistics Division, Rome, Italy
[3] Columbia University, Centre on Global Energy Policy, New York, USA
[4] United Nations Statistics Division, New York, USA
[5] International Energy Agency, Paris, France
[6] University of Gastronomic Sciences, Bra, Italy
[7] Massey University, Palmerston North, New Zealand
*Corresponding author:* Francesco N. Tubiello, francesco.tubiello@fao.org
**Abstract.** Fossil-fuel based energy use in agriculture leads to $CO_2$ and non-$CO_2$ emissions. We focus on emissions
generated within the farm gate and from fisheries, providing information relative to the period 1970-2019 for both
energy use as input activity data and the associated greenhouse gas (GHG) emissions. Country-level information
is generated from UNSD and IEA data on energy in agriculture, forestry and fishing, relative to use of: gas/diesel
oil, motor gasoline, liquefied petroleum gas (LPG), natural gas, fuel oil and coal. Electricity used within the farm
gate is also quantified, while recognizing that the associated emissions are generated elsewhere. We find that in
2019, annual emissions from energy use in agriculture were about 523 million tonnes (Mt $CO_{2eq}$ yr$^{-1}$), while
including electricity they were 1,029 Mt $CO_{2eq}$ yr$^{-1}$, having increased 7% from 1990. The largest emission increases
from on-farm fuel combustion were from LPG (32%), whereas significant decreases were observed for coal (-
55%), natural gas (-50%), motor gasoline (-42%) and fuel oil (-37%). Conversely, use of electricity and the
associated indirect emissions increased three-fold over the 1990-2019 period, thus becoming the largest emission
source from energy use in agriculture since 2005. Overall the global trends were a result of counterbalancing
effects: marked decreases in developed countries in 2019 compared to 1990 (-273 Mt $CO_2eq$ yr$^{-1}$) were masked by
slightly larger increases in developing and emerging economies (+ 339 Mt $CO_2$ eq yr$^{-1}$). The information used in
this work is available as open data at: https://zenodo.org/record/5153241 (Tubiello and Pan, 2021). The relevant
FAOSTAT (FAO, 2021) emissions database is maintained and updated annually by FAO.

## 31 1. Introduction

Agricultural production more than doubled over the period 1990-2019, with additional increases of more than 50%
expected to 2050, to meet projected increases in food demand (FAO, 2018; Calicioglu et al., 2019). Historically,
productivity increases were achieved through transitions from traditional, extensive agri-food systems to modern,
intensive production systems, characterized by greater energy use within the farm (Smil, 2008). Direct on-farm
energy inputs include fuel to power tractors and other agricultural field machinery, irrigation pumps, heat to warm
greenhouses and animal shelters. Other uses beyond the farm may include power for forestry machinery and fishing
vessels. We consider herein additionally the energy used to generate electricity that may be used on the farm in



substitution of on-site fossil-fuel combustion, but do not include all other indirect energy use that is typically
addressed in life-cycle type analyses (FAO, 2011; Sims et al. 2015; FAO, 2018).
On-farm energy use is a significant component of agricultural production and growth (Utz, 2011), however it often
escapes analyses of greenhouse gas (GHG) emissions in agriculture. Indeed, the 'agriculture' sector within national
GHG inventories (NGHGI), which countries submit regularly to the UN Framework Convention on Climate
Change (UNFCCC), contains only non-$CO_2$ emissions from crop and livestock bio-physical processes, for instance
enteric fermentation in ruminants or nitrous oxide from fertilizers on cropland (IPCC, 2006; Tubiello et al., 2019).
The on-farm energy use emissions are reported instead under the 'Energy' sector of the NGHGI, therefore often
escaping attention in food-related emissions analysis relevant to National Determined Contributions (Tubiello et
al., 2021). Within the UNFCCC context, emissions from agriculture are currently about 5 Gt $CO_2$eq $yr^{-1}$, having
increased by roughly 50% since 1961 (Tubiello, 2019). They are dominated by livestock processes, and are fairly
equally split between $CH_4$ and $N_2O$ components, respectively in single gases units corresponding to annual
emissions in 2018 of 140 Mt $CH_4$ $yr^{-1}$ and 7.7 Mt $N_2O$ $yr^{-1}$ (FAO, 2020; Tubiello et al., 2021).
Energy use in agriculture, forestry and fisheries nonetheless deserves more attention than paid in current reporting
and associated studies, because it is an important food production component deserving analysis in its own right
alongside the biophysical crop and livestock processes mentioned above. Additionally, it offers significant
opportunities for on-farm mitigation actions directly focussed on $CO_2$ (Dyer et al., 2014). This paper therefore
focuses on quantifying the GHG emissions that arise from the combustion of fossil fuels for energy use in
agriculture, forestry and fisheries (capture fishing and aquaculture). As detailed in the methods section, our
quantification will focus mostly on the farm and on fishing activities, assuming that emissions associated to energy
used in forestry is negligible—i.e., it will focus on energy use for farm operations, for aquaculture and for powering
fishing vessels. We include additional estimates of the emissions associated to the off-site generation of electricity
used on the farm, tracking results both separately for electricity and on-site fossil fuel use, as well as in the
aggregate.
Information on energy consumption in different agricultural operations is available from the literature, albeit there
is a lack of consistent global data with country detail provided over relevant time series. Available information
indicates that in-farm energy demand in OECD countries is mainly for crop cultivation, harvesting, heating
protected crops in greenhouses, crop drying and storage, water pumping and livestock housing (OECD, 2008).
Furthermore, on-farm use in high-GDP countries (20 GJ/ha) is almost double the use in low-GDP countries (11
GJ/ha) (FAO, 2011a). Fossil fuel energy inputs have reduced labor inputs, or around 152 MJ for every man-hour
of labor inputs in high-GDP countries, and 4 MJ in low-GDP countries (Sims, 2014).
Smil (2008) and FAO (2011) estimated global direct and indirect energy use in agriculture in the early 2000s at 17
EJ, of which 5 EJ to power machinery; 4 EJ for animal husbandry, aquaculture, and fisheries; 2 EJ to produce and
maintain agricultural machinery; 5 EJ to extract, synthesize and distribute fertilizers; 0.5 EJ to manufacture
pesticides and herbicides; and 0.3 to manufacture irrigation systems. Hence direct energy use in agriculture was a
bit more than half this total, about 9 EJ. In addition to these amounts, energy use in agriculture includes electricity
from the grid, decentralized renewable sources including bioenergy, conventional technologies, mechanical and
thermal energy and biodiesel/biofuels. In many traditional systems, human labour and draught animal power add
significant energy inputs.





## 2. Materials and methods

Data on energy use in agriculture forestry and fisheries, by fuel type, over the annual time series 1990-2019, were available from UNSD and IEA. These Agencies regularly collect energy data from member countries, including for use in agriculture, forestry and fishing. Biofuels, renewables, and other energy carriers derived from biomass, were analyzed but not considered for calculating GHG emissions, since they were assumed to be carbon neutral (IPCC, 2006). Energy use data from the UNSD Energy Statistics Database (UNSD, 2020) included the following fuels, over the period 1970-2019: Diesel oil; Motor gasoline; Liquefied petroleum gas (LPG); Natural gas, including Liquefied Natural Gas (LNG); Fuel oil; Hard coal. Electricity use data were also taken from the same database.

### 2.2 Gap filling

The information used in this work is available as open data at: https://zenodo.org/record/5153241 (Tubiello and Pan, 2021). The relevant FAOSTAT (FAO, 2021) emissions database is maintained and updated annually by FAO.

### 2.3 Gap filling

The energy use data sourced from UNSD were gap filled for both improving the quality of available time series by country and generating data for missing countries. The original set had several missing data points especially for Africa (FAO, 2021). First, a simple linear gap-filling method was applied to estimate data points missing within intervals with data points, over the time period 1970-2019. Conversely, gap-filling of values for carrying backward and forward values without an available interval was performed by applying sub-regional trends. Finally, time series for countries with no data were generated with a multivariate approach, i.e., by computing the sub-regional energy use in agriculture divided by the sub-regional total energy use, and applying the coefficient to the time series of national total energy use, which was available in the UNSD database without major gaps. We validated our gap-filling method by performing random substitutions of existing values and computing the associated error, which was on average below 5%.

### 2.4 Emissions Estimates

The activity data on energy use described in previous sections served as input for estimates of GHG emissions, made following the Tier 1 method of the Guidelines of the Intergovernmental Panel on Climate Change (IPCC, 2006). In particular, we used default fuel-specific $CO_2$ emission factors for off-road mobile combustion sources (e.g., tractors, harvesters and other mobile machinery) and stationary combustion sources (i.e., irrigation pumps, space heating), within the following formula:

$$E_i = AD_i * EF_i$$

Where $E_i$ are the emissions (in t $CO_2$ $yr^{-1}$) for energy carrier $i$, computed by multiplying the amount of fossil fuel type $AD_i$ (GJ $yr^{-1}$) by the relevant emission factor $EF_i$ (t $CO_2$ $GJ^{-1}$). The default emission factors applied to relevant fuel categories were those for stationary combustion in the residential and agriculture/forestry/fishing farms



categories, assumed by IPCC to be used for power generation (heat and/or electricity) (Tab. 2). Fuels reported in
metric tons were converted to GJ by assuming a net calorific value of 43.0 GJ/t for diesel, 44.3 GJ/t for gasoline,
47.3 GJ/t for LPG, 44.2 Gg/t for natural gas liquids, 40.4 GJ/t for fuel oil, 25.8 for coal[1] (IPCC, 2016).
Finally, country-specific grid emission factors needed to estimate $CO_2$ emissions from electricity used were taken
from IEA. The associated emissions of $CH_4$ and $N_2O$ were not considered, as our calculations (not shown)
indicated the latter would be five to six orders of magnitude smaller compared to $CO_2$, on a per ton basis.
Emissions from fisheries were estimated as a separate item (until 2018), using dedicated IEA data, and for
information purposes only, i.e., they were assumed to represent additional information, since energy used in
agriculture, forestry and fisheries are already included in the UNSD energy statistics. Fisheries statistics from IEA
were limited to OECD countries. Only diesel and fuel oil for powering fishing vessels and aquaculture are reported
under fisheries, since these two fuels (followed by heat) represent the bulk of energy used in the sector.
Uncertainties were derived by applying ranges for GHG emission factors provided by IPCC 2006 to fuels
considered and an error of 5% for emissions associated with electricity consumption (calculated based on the global
energy mix for electricity generation in the IEA database).
**2.3 Limitations and uncertainty**
There are limitations and uncertainties associated with the estimates presented herein. First, we note that the input
data on energy refers to use in agriculture, forestry and fisheries, without further breakdown. While we refer often
to the associated emissions as generated within the farm gate, they include components of unknown relative
magnitude that are in fact generated through forestry and fisheries activities. For the latter, we have provided a
partial and incomplete breakdown in the database, using IEA fisheries data. Second, the underlying data on energy
use have significant geographical gaps, especially in Africa, as well as temporal gaps, particularly before 1990.
For estimates of GHG emissions, we applied default IPCC methods and uncertainty values for EFs to compute the
error propagation in equation (1) above, finding an uncertainty range in emissions of -7 to 16% (Figs. 4-5).
**2.4 Data availability**
The GHG emission data presented herein cover the period 1990-2019, at the country level, with regional and global
aggregates. Significant gaps in some countries and regions, especially Africa, imply that specific regional estimates
may be systematically underestimated. Additionally, statistics on energy consumption and emissions from fisheries
are highly uncertain and likely underestimates, considering that significant amounts of fuel consumed by small
vessels, constituting a majority of the global fishing fleet, are not typically reported in official statistics.
Data on energy use in agriculture and associated emissions used in this work are available as open data at:
https://zenodo.org/record/5153241 (Tubiello and Pan, 2021). The relevant FAOSTAT (FAO, 2021) database is
maintained and updated annually by FAO.

3   **Results**
Our estimates indicated that world-total GHG emissions from energy use in agriculture including electricity were
above 1 billion tonnes in 2019 (1,029 Mt $CO_2$eq yr$^{-1}$; 7% greater than in 1990). The average annual increase was

---

[1] We assumed that coal used in agriculture is mostly 'bituminous coal.'



0.2% over the period 1990-2019 and was consistent with the overall growth in agricultural emissions within the
farm gate. Almost half of the estimated emissions (496 Mt $CO_2$eq $yr^{-1}$) arose from combustion of fossil fuels for
power generation of electricity used on the farm. The most important energy sources after electricity were
gas/diesel oil and coal, while motor gasoline, typically associated to field machinery and tractors use in developing
countries, contributed a mere 5% of the total (Figs. 2-3). Emissions from electricity grew rapidly over the study
period (mean annual growth rates of more than 6%), overtaking gas diesel oil and motor gasoline as the main
emission source by roughly the year 2012. This, together with an increase of LPG use, suggests a global transition
towards cleaner on-farm energy use, considering grid electricity is typically associated to lower emissions per
energy compared to single fossil fuel sources. At the same time, use and hence emissions from natural gas, fuel oil
and coal were rather constant over the period 1990-2019, about 38, 123,  and 25  Mt $CO_2$eq $yr^{-1}$ on average. While
data for on farm energy use were rich in coverage, trends in emissions from use of diesel oil and fuel oil in fishing
vessels were limited by data paucity. Within such limitations, we find a small, decreasing share of emissions from
fishing vessels compared to world-total energy use in agriculture, with a total contribution in 2018 (the breakdown
of energy used in fisheries is available only until 2018) of about 27 Mt $CO_2$eq $yr^{-1}$ (3%).

**3.2 Regional Distributions and Trends**

Our results indicate that on-farm energy use is an important and increasing component of GHG emissions in
agriculture (Fig. 1). Emissions declined in Annex I countries over the period 1990-2019, especially energy from
coal (-88%) and fuel oil (-77%). Such decline was more than counterbalanced by increases in energy use in non-
Annex I parties (NAI), with significant increases in emissions from electricity (three-fold increases since 1990)
(Fig. 4).  Asia and Europe were the largest emitters among FAO regions, although with starkly different trends
over 1990-2019. Indeed, while emissions in Europe decreased over the whole period, from 730 Mt $CO_2$eq $yr^{-1}$ in
1970 to 410 Mt $CO_2$eq $yr^{-1}$in 1990, and further decreased to 145 Mt $CO_2$eq $yr^{-1}$ in 2019, emissions in Asia nearly
doubled over 1990 to 2019, from 380 Mt $CO_2$eq $yr^{-1}$ to 629 Mt $CO_2$eq $yr^{-1}$, while they were 453 Mt $CO_2$eq $yr^{-1}$ in
1970.  Africa was a significant emission source in 2019, having more than doubled since 1990, from 18 Mt$CO_2$-
eq to 48 Mt $CO_2$eq $yr^{-1}$. Emissions increased more than 55% in Latin America, but only 18% in North America.
The smallest contributor to global emissions was Oceania, despite increases by nearly 55% from 1990 (Fig. 7).
Top emitting countries in 2019 in terms of energy use in agriculture were China (233 Mt $CO_2$eq $yr^{-1}$), followed by
India (212 Mt $CO_2$eq $yr^{-1}$) and the USA (79 Mt $CO_2$eq $yr^{-1}$). The top 10 emitting countries were responsible for
nearly two-thirds of the world total (Fig. 8).

**3.3 Indicators**

We developed indicators by cropland area and agricultural production value to help us disentangle effects of
country agricultural size, both in terms of area and economy. We defined GHG emission intensity per unit cropland
as the total GHG emissions from energy use in agriculture divided by total cropland area of a country. Likewise,
energy GHG intensity per production value was computed by dividing total GHG from national energy use in
agriculture by total agricultural value added. Data for denominators of both indicators were taken from FAOSTAT
(FAO, 2021b, c).
Our results indicate that energy GHG emissions per unit cropland have been fluctuating but have been substantially
stable over the last two decades. Nonetheless, significant differences can be noted among regions (Fig. 11). While




Europe has decreased significantly its energy-related GHG emission intensity in agriculture (-57%) in the period 1990-2018, Africa, Central America and Asia have increased it substantially (+88%, +51% and +44% respectively). This means that more GHG emissions are associated with the cultivation of one unit of cropland in these regions. In absolute terms, the lowest energy intensity per unit of cropland in 2018 was achieved in Africa (0.16 t $CO_2$eq ha$^{-1}$), followed by Oceania (0.38 t $CO_2$eq ha$^{-1}$), South America (0.42 t $CO_2$eq ha$^{-1}$) and Europe (0.48 t $CO_2$eq ha$^{-1}$).

In terms of energy-related GHG emissions to agricultural value added, the picture is substantially different, with Europe having significantly improved its energy intensity since 1990 (-68%), followed by Asia (-61%), Latin America and the Caribbean (-54%), Northern America (-53%) and Oceania (-45%), while Africa's intensity remained substantially stable over the last two decades.

In 2019, high levels of GHG emissions per capita (from energy used in agriculture) were estimated for Faro Islands, Greenland and Iceland (Fig 13). In those territories, emissions from gas/diesel oil take more than two-thirds of the total. Fishing is one of the most responsible factors contributing to the high per capita emission from energy use in agriculture in Faroe Island, as fishing vessels take almost one-third of energy use at national level. Fishing is also the primary industry in Iceland. For Greenland, fishing is the second-largest industry by employment. Though Greenland has the highest ratio of using renewable energy (70%), fishing remains a sector depending on traditional fossil fuels.

## 4   Discussion

Emissions from energy use in agriculture are only about one-fifth of the total in $CO_2$eq generated from crop and livestock production (Tubiello et al. 2019), however they represent an important contribution in terms of $CO_2$ gas, the other process emitting $CO_2$ on the farm being the drainage of organic soils. They are therefore of great importance to GHG mitigation in agriculture. In terms of comparing these results with the existing literature, we note that our approach covers only 7.2 of the 8-10 EJ usually estimated for total fuel consumption within the farm gate (Arizpe et al., 2011; FAO, 2011; Smil, 2008). Additionally, our estimates of energy use in fisheries is admittedly incomplete (0.3 EJ) compared to amounts reported in other studies (Buhaug et al., 2009; FAO, 2011). The reason is that we focused only on electricity and on the most relevant fuels consumed in agriculture, but not all. Specifically for fisheries, the relatively low coverage is also due to the fact that still few countries report disaggregated energy consumption statistics for fisheries alone.

Electricity generation and gas/diesel oil used in agriculture were the two most important emissions sources, responsible for roughly 40% of the total on average during the period 1990 -2019. Electricity is used for different agriculture purposes: irrigation, processes that require heat or mechanical power, such as drying or milling. LPG, natural gas, and heavy fuel oil are typically used for heat generation and, in some rare cases, for motive power. Apart from some sharp variation of their total consumption in agriculture between consecutive years, mainly at the beginning of the '90s, probably due to reporting issues of important consumer countries such as India and the dissolution of the USSR, their emissions remained relatively stable. Compared to other emissions, coal and fuel oil emissions decreased over the last few years, while agricultural production still increased. This can be explained by updated energy use structure - the increased uptake of cleaner energy carriers such as electricity and LPG over fuel oil and coal for heating. China, for example, one of the major emitting countries, decreased emissions from fuel oil use by 48%, while increased emissions due to diesel use by around 59 % and emissions due to electricity



use by over 170% over the same period 1990-2019. There is anyway still a long way to go to decrease emissions in the agricultural sector in China, due to its still very high reliance on coal as a heat source.

Unlike other regions, Europe's emissions went significantly down, partly because less energy was consumed by primary production in absolute terms. Also, Europe has gradually moved from high GHG emitting energy carriers such as coal and fuel oil towards cleaner ones, such as natural gas and electricity. This is confirmed by the additional analysis done using the energy-related GHG intensity indicators. This analysis shows how Europe has been steadily improving its agricultural GHG intensity (both in terms of unit of cropland and of unit of agricultural production value), thus providing a good example for other regions.

## 5 Conclusions

This paper provides details of a new dataset added to the existing section of FAOSTAT, which contains information about emissions due to agricultural activities, and which was just opened publicly online (July 2021). It also provides an analysis of energy-related GHG intensity in agriculture, per unit of cropland and per unit of agricultural production value, which has not been published yet. It complements the analysis with selected GHG emission intensity indicators, which are derived directly from FAOSTAT. The calculation makes use of official statistics as reported by countries to the UN, applying IPCC Tier 1 default emission factors for fuels and IEA country-specific emission factors for electricity generation (considering the national energy mix) and relies on official energy consumption in agriculture data reported by countries to the UNSD and the IEA. Further to the above, the share of emissions on fisheries' energy use is estimated and reported separately as a subset. These estimated emission shares provide references to their relevance compared with total emissions but should be used with relevant uncertainties taken into consideration.





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





1 TABLE LEGENDS



1    FIGURE LEGEND

28
29





|  | $CO_2$ | | | $CH_4$ | | | $N_2O$ | | |
|---|---|---|---|---|---|---|---|---|---|
|  | Default (kg/TJ) | Lower | Upper | Default (kg/TJ) | Lower | Upper | Default (kg/TJ) | Lower | Upper |
| Gas/Diesel oil | 74100 | 72600 | 74800 | 4.15 | 1.67 | 10.4 | 28.6 | 14.3 | 85.8 |
| Motor gasoline [2] | 69300 | 67500 | 73000 | 80 | 32 | 200 | 2 | 1 | 6 |

**Table 1. Fuel-specific emission factors for agriculture off-road mobile combustion sources and machinery applied (IPCC 2006)**

|  | $CO_2$ | | | $CH_4$ | | | $N_2O$ | | |
|---|---|---|---|---|---|---|---|---|---|
|  | Default (kg/TJ[3]) | Lower | Upper | Default (kg/TJ) | Lower | Upper | Default (kg/TJ) | Lower | Upper |
| Liquefied Petroleum Gases | 63100 | 61600 | 65600 | 5 | 1.5 | 15 | 0.1 | 0.03 | 0.3 |
| Natural gas | 56100 | 54300 | 58 300 | 5 | 1.5 | 1.5 | 0.1 | 0.03 | 0.3 |
| Residual fuel oil | 77400 | 75500 | 78800 | 10 | 3 | 30 | 0.6 | 0.2 | 2 |
| Other bituminous coal | 94600 | 89500 | 99700 | 300 | 100 | 900 | 1.5 | 0.5 | 5 |

**Table 2. Fuel-specific emission factors for stationary combustion in the residential and agriculture/forestry/fishing/fishing farms categories applied (IPCC 2006)**

---

[2] The default emission factors regard 4-stroke motor gasoline engines.
[3] kg of greenhouse gas per TJ on a Net Calorific Basis

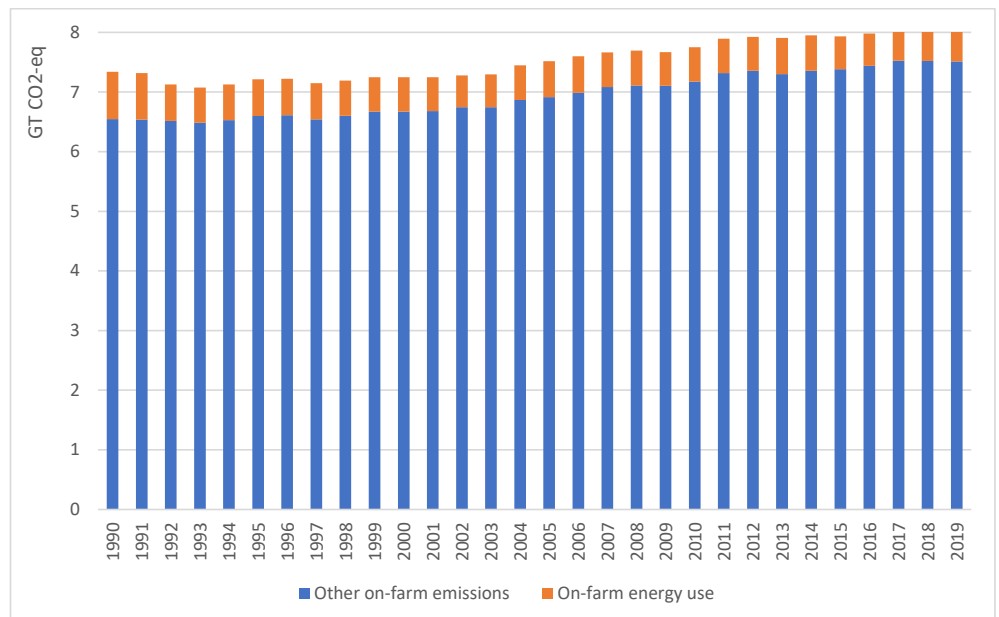

2  **Figure 1. Global emissions from energy use (orange bars) and emission from other on-farm total, excluding energy (blue**

3  **bars) per year (Gt CO2-eq). Source: FAOSTAT, based on data from IEA and UNSD, 2021**





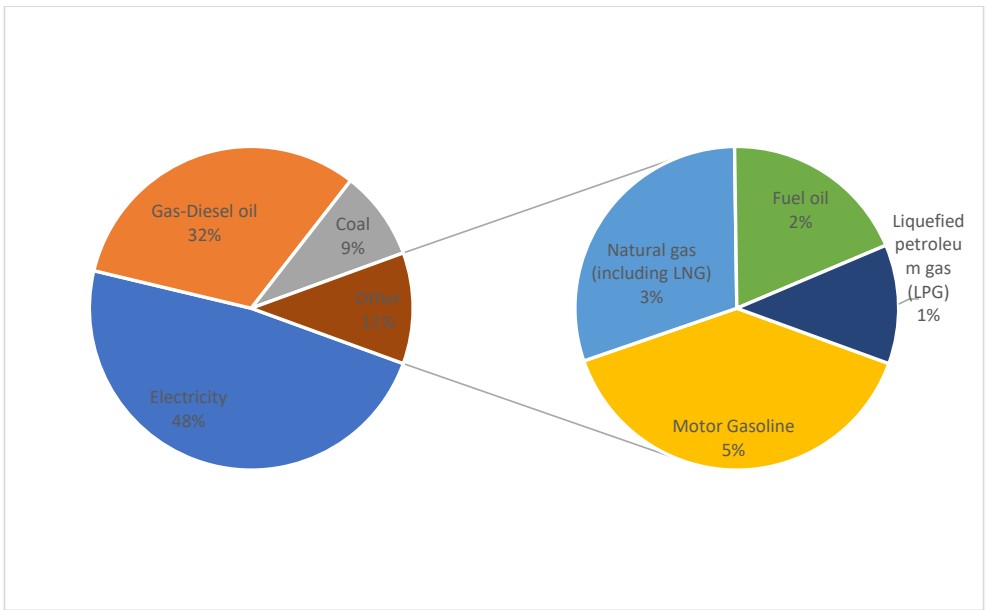

**Figure 2. Global shares of emissions due to energy use in agriculture in 2019, by energy carrier (CO2-eq). Source:**

**FAOSTAT based on data from IEA and UNSD, 2021**



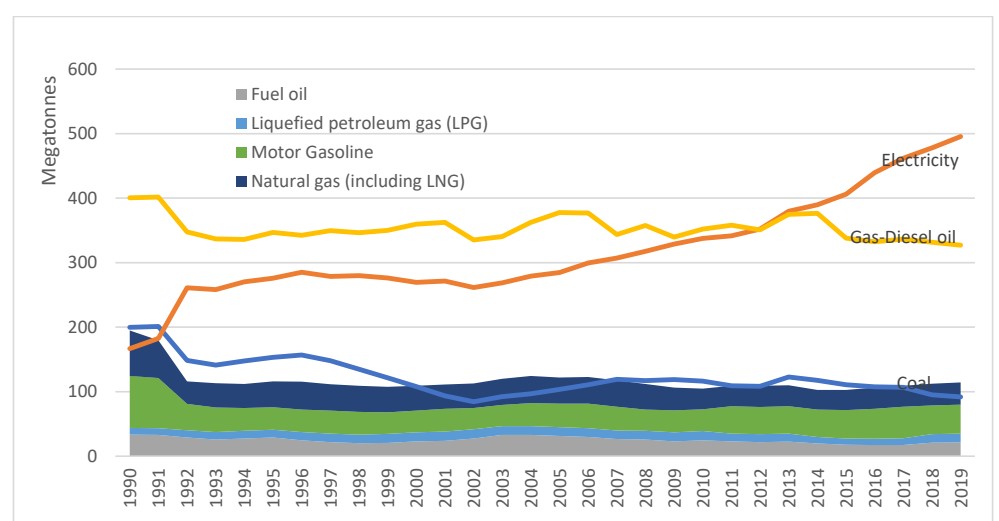

3  **Figure 3. Global GHG emissions from energy use in agriculture from 1990 to 2019, by energy carrier (MtCO2-eq).**

4  **Source: FAOSTAT, based on data from IEA and UNSD, 2021**



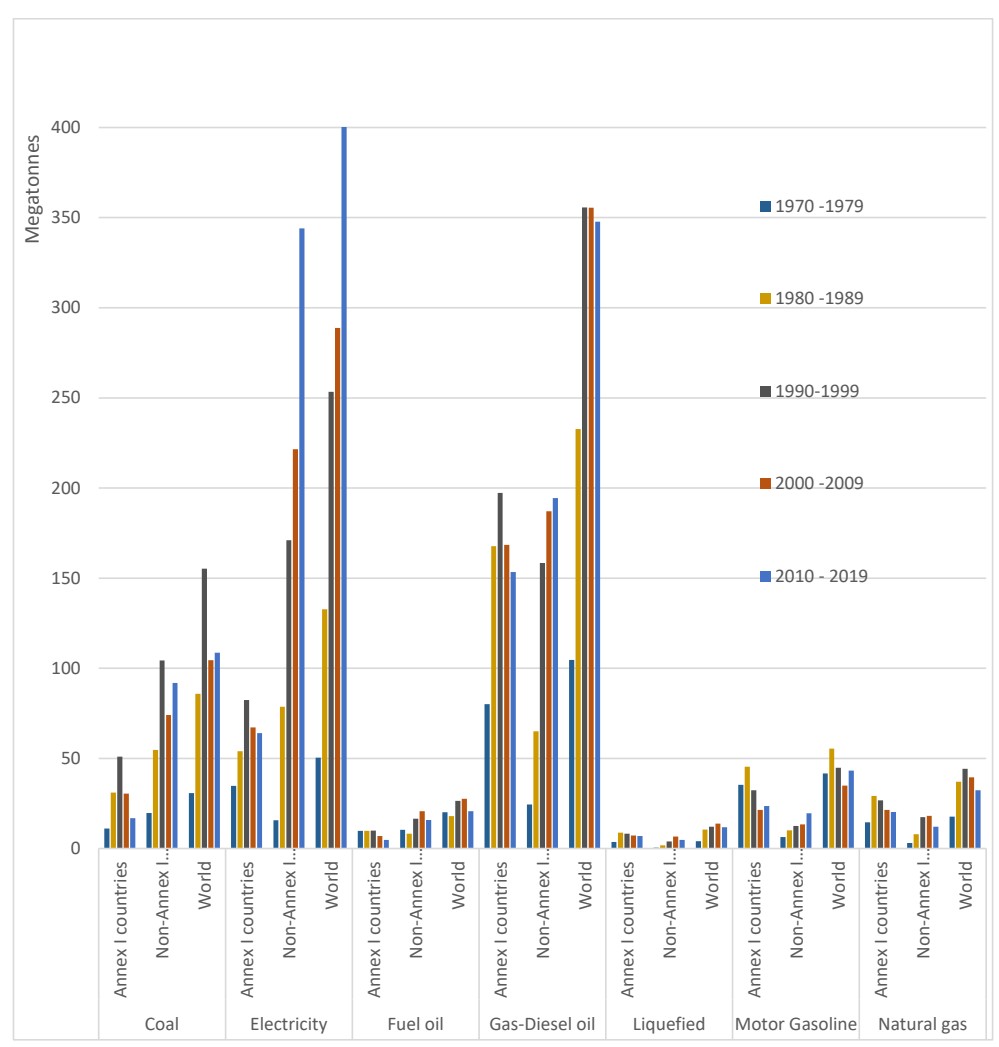

2    **Figure 4. GHG emission trends from 1990 to 2019 for Annex I and NAI, World, by energy carrier (MtCO2-eq). Source:**

3    **FAOSTAT, based on data from IEA and UNSD, 2021**



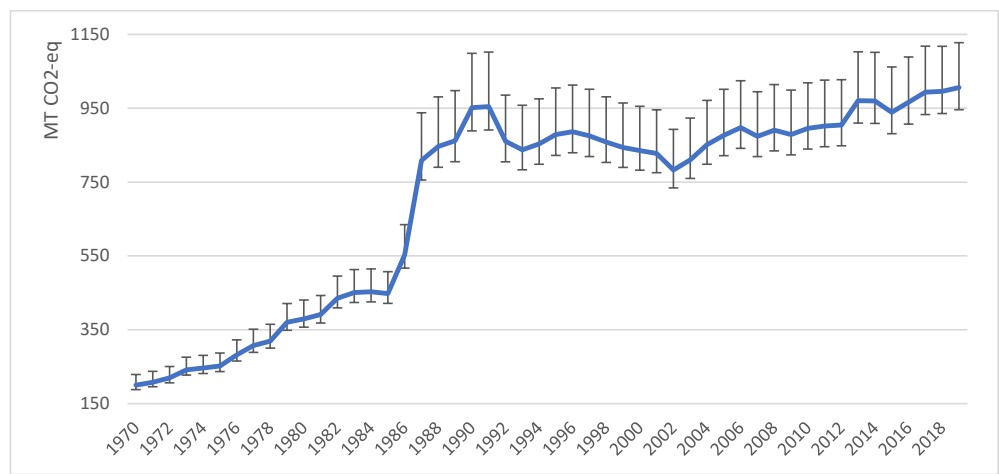

**Figure 5. Trend in global GHG emissions from 1990 to 2019, with uncertainty ranges (MtCO2-eq). Source: FAOSTAT, based on data from IEA and UNSD, 2021**





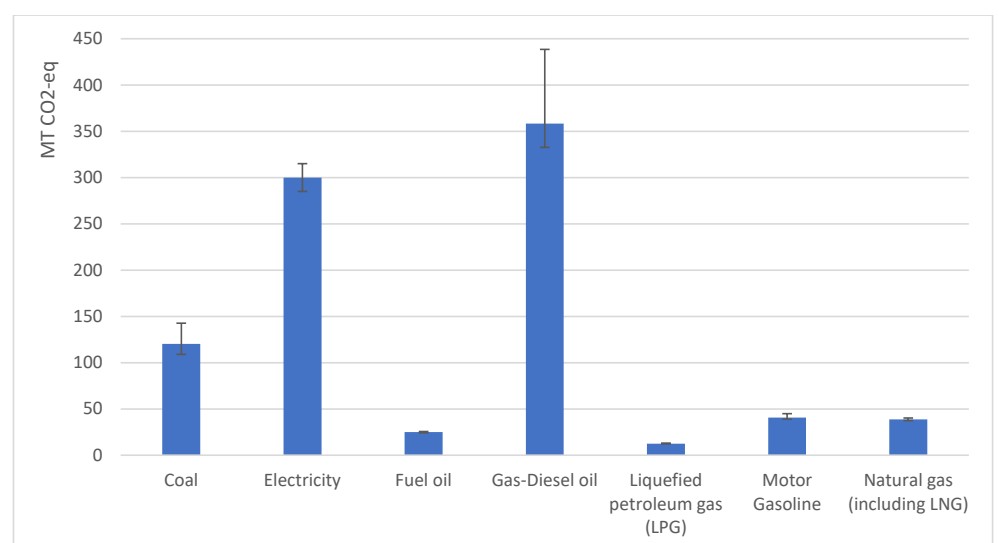

**Figure 6. Global GHG emissions from energy use in agriculture (average 1990 – 2019) by energy source with uncertainty ranges (MtCO2-eq). Source: FAOSTAT, based on data from IEA and UNSD, 2021**



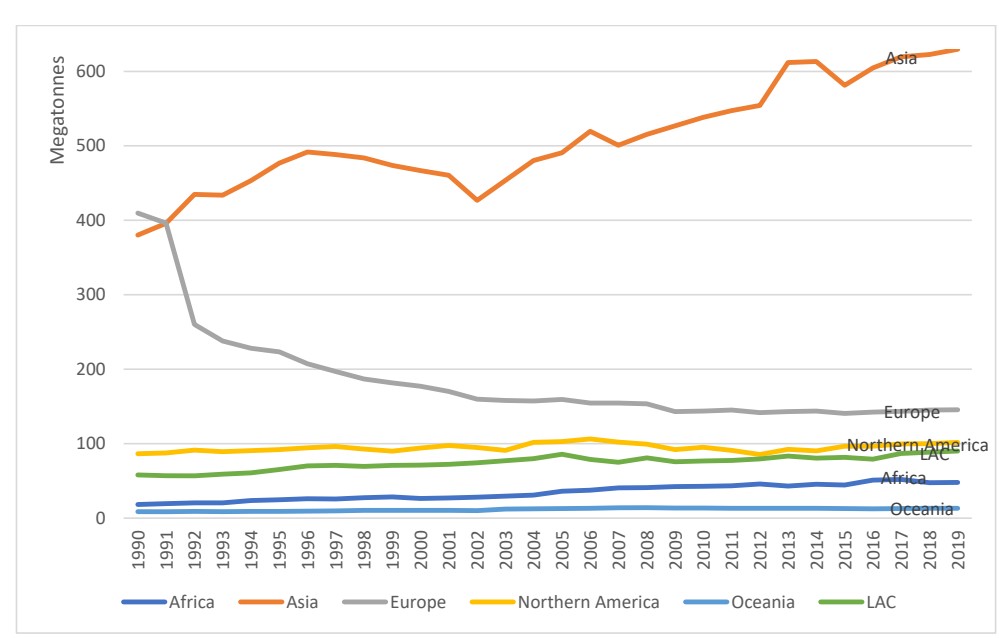

3 **Figure 7. GHG emissions from energy use in agriculture from 1990 to 2019, by region (MtCO2-eq). Source: FAOSTAT,**

4 **based on data from IEA and UNSD, 2021**



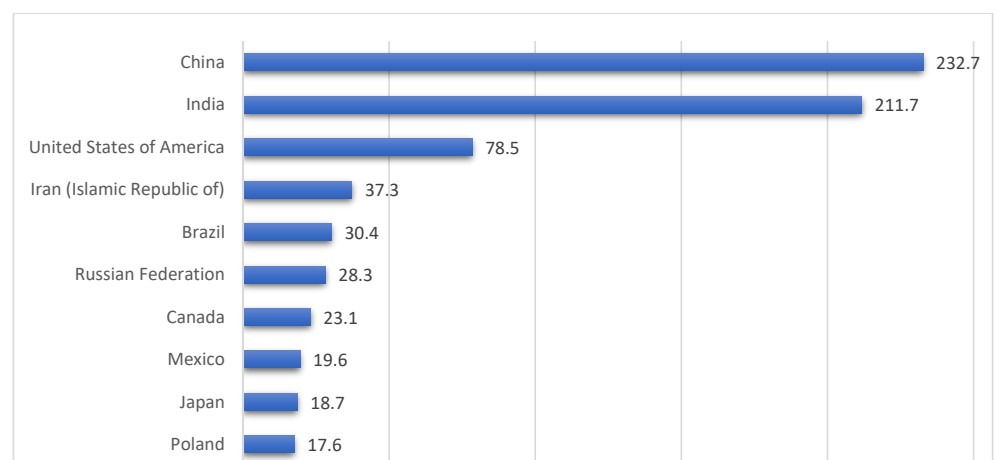

**Figure 8. Top 10 countries emitting GHG from energy used in agriculture in 2019 (MtCO2-eq). Source: FAOSTAT, based on data from IEA and UNSD, 2021**

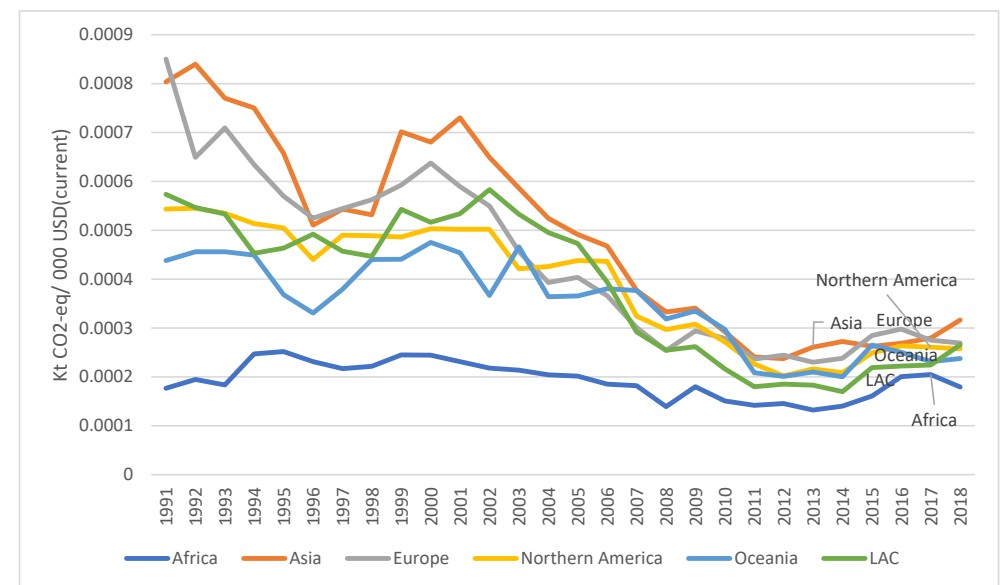

**Figure 9. GHG emission from energy use in agriculture per gross agriculture production value 1991-2018**

**(KtCO2eq/1,000 current USD)**

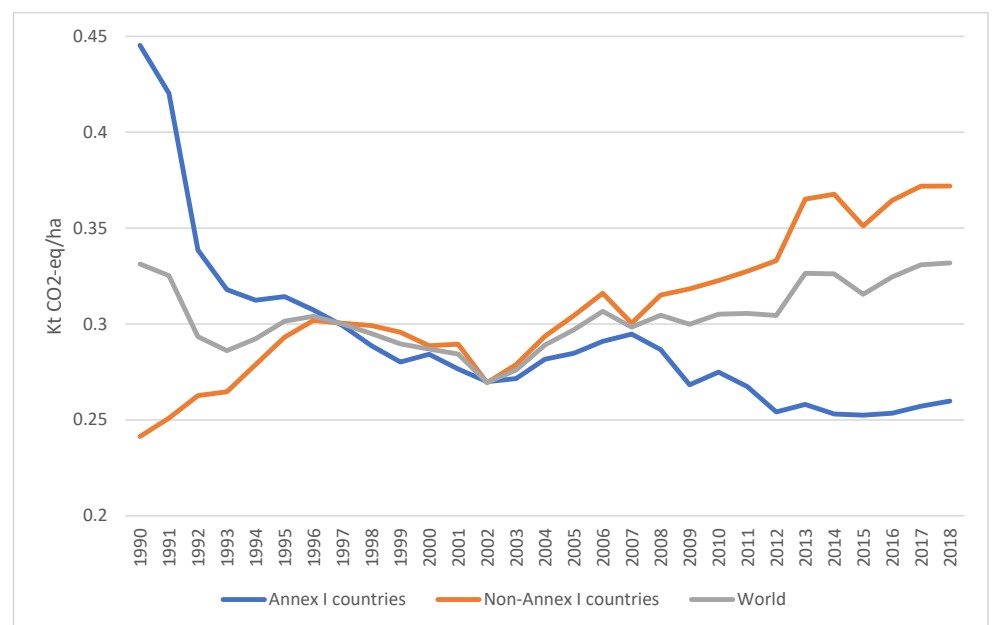

2  **Figure 10. GHG emission from energy use in agriculture per unit of cropland for Annex I and Non-Annex I countries**

3  **1990-2018 (Kt CO2-eq/ha). Source: FAOSTAT, based on data from IEA and UNSD, 2021**



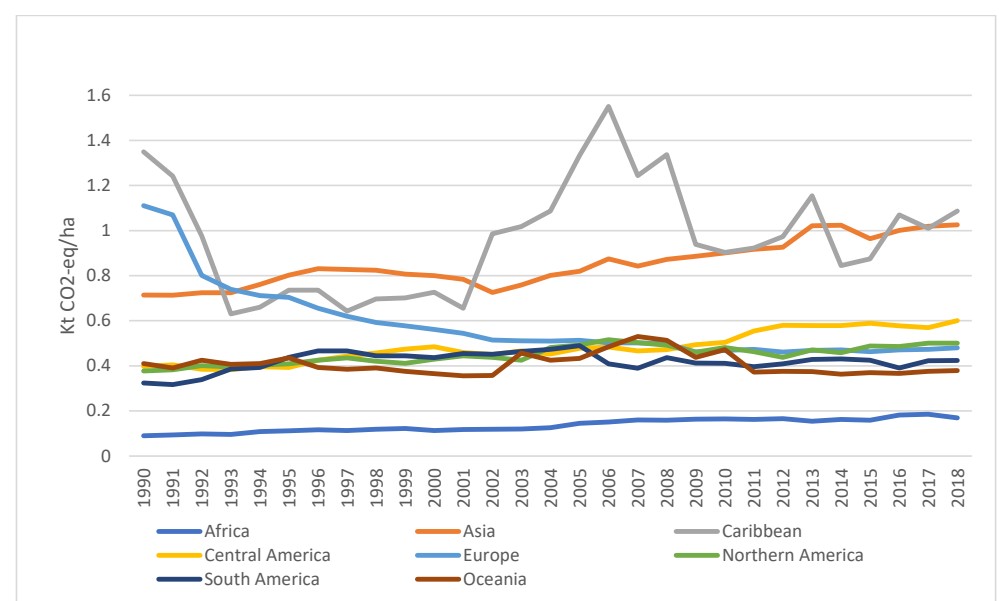

3  **Figure 11. GHG emission intensity trends per unit of cropland by region (Kt CO2-eq /ha). Source: FAOSTAT, based**

4  **on data from IEA and UNSD, 2021**

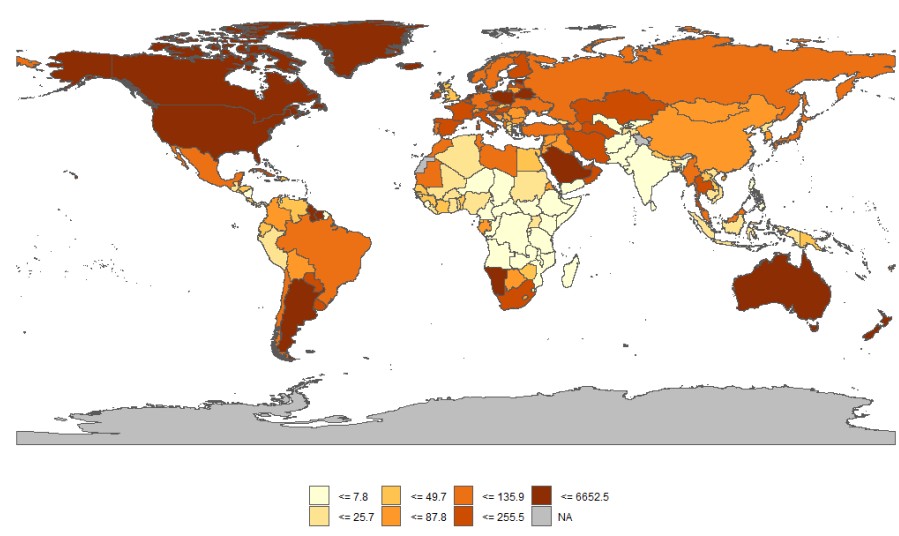

2 **Figure 12. GHG emission from energy used in agriculture per capita in 2019 (Kg CO2-eq /person)** [4]

---

[4] Energy data from FAOSTAT, 2021. Population data from the World Bank
(https://data.worldbank.org/indicator/SP.POP.TOTL), with some countries from United Nations, Department of Economic
and Social Affairs, Population Division. Falkland Islands (Malvinas), Guadeloupe, French Guyana, Martinique, Niue,
Réunion, Romania, Palestine, Democratic Republic of the Congo.



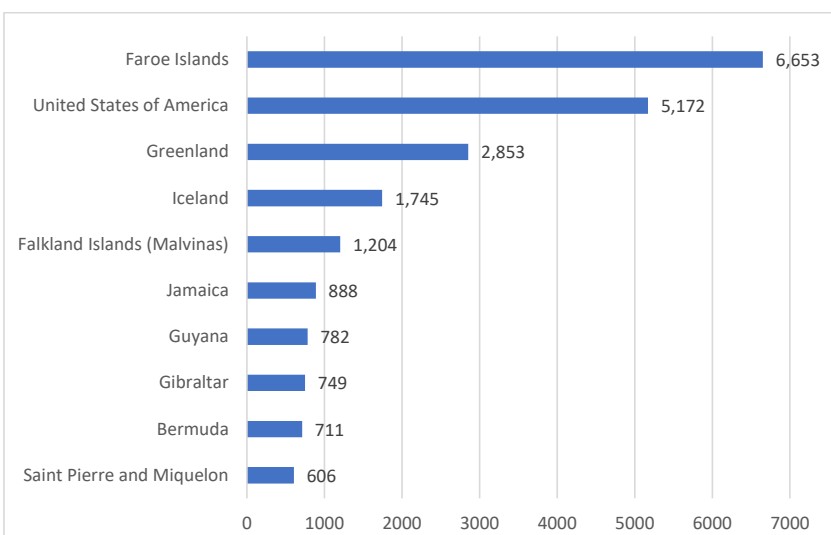

2 **Figure 13. Top 10 GHG emitting countries from energy use in agriculture per capita in 2019 (Kg CO2-eq/person)**

