# Peer review of "Emissions of greenhouse gases from energy use in agriculture, 1"

_Earth System Science Data, 2021_

## Author Comment (AC1)

The authors reconstructed a country-level energy-related GHG emission inventory in the agriculture, forestry and fisheries sector from 1970 to 2019. The work is worthy of recognition. However, there are still many aspects that need to be improved. The following are my comments. A substantial revision is needed.

1. In the Introduction section, it is recommended that the description of previous research on energy-related GHG emissions in the agriculture, forestry and fisheries sector (on Page 2, Line 25-39) be placed before what this research has done (Page 2, Line 18-24). At the same time, the narrative logic of this part needs to be improved accordingly.

**ANSWER 1**. Thank you for your comment. We have adjusted the introduction section accordingly, providing at first the context and information on previous research before presenting the paper.

2. On page 2, Line 32-36, the description of the two literature (Smil (2008) and FAO (2011)) seems too detailed, resulting in messy data and not easy to understand. It is recommended to summarize and show only the most important content.

**ANSWER 2**. We re-organized the text on page 2 as suggested, by placing the context information in one place within the text; and by keeping the information on context and on energy consumption of different food system components, albeit in a more streamlined manner.

3. In the Materials and methods section, the first paragraph (on Page 3, Line 2-9) mentioned that the basic data sources of this study are UNSD and IEA. But the authors used a lot of space to describe UNSD dataset, and did not mention anything in terms of IEA.

**ANSWER 3** We added some text to clarify the use of both UNSD and IEA data within the manuscript. We make it now clearer that the analysis builds almost entirely on UNSD data--and this is why it is described more extensively. Conversely, IEA data were used only for the country grid emission factors and to provide a breakdown for fisheries.

4. In the Materials and methods section, the subtitles are disorderly, the paragraphs are chaotically divided, and the text descriptions are repeated. It is recommended to make in-depth revisions.

**ANSWER 4**. We apologize for this and thank the reviewer for his patience. The text has been thoroughly revised for increased readability.

5. On Page 4, Line 5-6, "The associated emissions of CH4 and N2O were not considered, as our calculations (not shown) indicated the latter would be five to six orders of magnitude smaller compared to CO2, on a per ton basis." This sentence needs some evidence to support it.

**ANSWER 5**. This paragraph has been reworded to clarify how non-CO2 electricity emissions were estimated, using that text to also better quantify the much smaller contribution of calculations of the CH4 and N2O components from electricity since it is quite complicated and, at the end, it is non-CO2 emissions are very small compared to CO2 (less than 5% of the total).

6.  As a data article, the uncertainty and technical verification part (Page 4, Line 16-24) seems too thin and needs to be improved drastically.

**ANSWER 6**. The uncertainty in our estimates is a propagation of uncertainty in activity data and emission factors. We acknowledge that the original manuscript limited this discussion to the emission. We have revised this section to adopt the tier1 approach of the IPCC Guidelines ot compute propagation of uncertainty across activity data and energy consumption data. Finally, we also better highlight an additional source of uncertainty outside the IPCC methods, due to the imputation methods of the input energy database form UNSD that is used in our work.

7.  The content of the results section is generally too simple, just briefly talk about the trend and a few simple indexes needs improvement.

**ANSWER 7**. We expanded the results section with expanded discussions at country and regional level.

8.  There are too many Figures (up to 13), and a lot of information is repeated, which needs to be improved.

**ANSWER 8**. We generated significant amount of information in this work, which could be well described through graphs. Nonetheless we value the reviewer's suggestion and limited the number of graphs to nine.

9.  There are some typos and simple grammatical errors in the manuscript, please check carefully.

**ANSWER 9**. We checked the language of the revised manuscript with the help of mother-tongue colleagues to help us correct for grammatical errors.

*Online reviewer 2*
https://essd.copernicus.org/preprints/essd-2021-262/#discussion

The manuscript presents a database of country-specific energy-related GHGs emission inventory for the agriculture, forestry and fisheries sectors from 1970 to 2019. It provides the country-specific information on the GHGs data and could potentially improve the understanding of long-term temporal trend of GHG emissions for the three sectors. To satisfy the needs of readers and potential users from multiple research background, the description of the data should be improved. In particular, I feel the data organization is not clearly explained and the details of the datasets should be better described. Moreover, the figures and tables should be improved as well. A substantial revision on the format of manuscript (including figures and tables) are needed.

1.  The figures and tables in the manuscript are not in good format. Should be carefully edited. Some figures actually provided very limited information and thus should be merged or reorganized (for example, Figures 6, 8 and 13).

**ANSWER 1**. Thanks for the suggestion. We double checked graphs and units and we reduced the number of graphs (from 13 to 9), keeping only the most significant ones.

2. The text needs improvement as well. Section 2.1 is missing and there are two Sections of Gap filling. Sections 2.3 and 2.4 repeated twice. Please be careful on the text.

**ANSWER 2**. The headings of the sections have been fixed. It was a mistake. Thank you for pointing it out.

3. I feel the description on the datasets (provided as .csv files) is insufficient. The datasets are valuable thus need better introduction. Why were those data fields selected and how were the data organized? What kinds of information could the dataset provide and how can the information be obtained? I encourage the authors to provide more examples.

**ANSWER 3**. Good point. We agree that the dataset is not self-explanatory as it is. We added a reference to the relevant FAOSTAT webpage with all information, including the metadata sheet with more detailed technical information on the dataset.

4. In Introduction, the structure of the source categories should be given clearly. The emissions of subcategories for the three sectors are important.

**ANSWER 4**. We think there is a misunderstanding on the scope of our analysis, which applies to the 'agriculture' sector, which includes forestry and fisheries. We clarified it in the introduction.

5. The explanations of the results are too simple. Some specific issues should be included or improved, for example, the long-term temporal trend and its driving factors; the spatial pattern of GHG emissions and its reason; and the species characteristics of emissions. Better explanation would improve the significance of the work.

**ANSWER 5**. As suggested, we added more details on the temporal trends, special patterns etc, especially in the results section. At the same time we note that the main objective of this paper is to present results and key trends of a new database, in itself a major innovation, using as it does country official data extensively and with enhanced granularity.

6. Could the authors indicate the fractions of emissions for the sectors concerned in this work to the total GHGs emissions by country?

**ANSWER 6**. As noted earlier, we think there is a misunderstanding: agriculture in this analysis includes all agricultural sub-sectors, including forestry and fisheries (as per FAO definition). It is now clarified in the text.

7. How are the data obtained in this work for the three sectors compared with other available studies including the IPCC estimates? A more comprehensive comparison should be conducted, if possible, at the country level.

**ANSWER 7**. According to IPCC guidelines, energy emission factors are covered under the sector "RESIDENTIAL AND AGRICULTURE/FORESTRY/FISHING/FISHING FARMS CATEGORIES", which includes agriculture, forestry and fisheries. All our analysis is done at the country level, and it is subsequently aggregated at different levels. We revised the text for increased clarity to this end.

---

## Author Response (AR2)

**Reviewer's comment:**

The author has made a lot of improvements according to the refs' comments. There are a few more minor comments and suggestions.

1. Although the author has limited the number of graphs, there are still too many figures. Is it possible to merge some graphics with similar information and display them in the form of sub-graphics?

**Answer:**

We merged some figures and reduced their number to 7 (from 13 figures of the first submission and 9 of the second submission). We merged the graphs on GHG from energy use per unit of cropland of Annex I/NAI and continents. They are now side-by-side, so that the Annex I/NAI trend uses another scale and is still clearly visible. Total energy emission trends by continent have been merged with top emitting countries, since both measure GHG from all energy carriers used in agriculture.

**Reviewer's comment:**

2. On page 2, Line 35-36, the authors try to use one single 'agriculture' sector to cover three sub-sectors: agriculture, forestry, and fisheries. It seems fine. However, in many other places in the paper, the 'agriculture, forestry and fisheries' is still used to describe the aggregated sector, which seems a bit confusing. No matter what kind of narrative, please be consistent.

**Answer:**

We revised accordingly, so that starting from page 2, where it is clarified that in our analysis "agriculture" includes the forestry and fisheries sub-sectors, the manuscript text always refers to "agriculture". We nonetheless added ''(including forestry and fisheries)'' whenever we felt it was important as a reminder to the reader.

**Reviewer's comment:**

3. According to my knowledge, IPCC estimated that the uncertainty for countries with less well-developed energy statistic systems may be on the order of ±10%, whereas the range for the countries with good energy collection systems is ±5% (this figure is also referred in this paper). The author found "an uncertainty range in emissions of -7 to 16%. The overall resulting uncertainty ranges are between -9 to +17%" (on page 5). How to explain why this upper bound data is so large? Need some explanation.

Yes, uncertainty in activity data from "well developed statistical systems" is ±5% and from "less developed statistical systems" is ±10% or more. We assumed an error of ±5% for activity data (also confirmed by UNSD Energy Statistics colleagues) since usually fossil fuel consumption statistics are very reliable as they are also used for taxation purposes. The -7 to 16% range represents the error associated with the IPCC default emission factors (by calculating upper and lower bound of emission factors of each fuel and taking a weighted average based on world fuel consumption data. The final resulting uncertainty range is the result of error propagation of activity data and emissions factors data, computed as per the default approach of the IPCC guidelines We clarified it in the text.

**Reviewer's comment:**

4. On page 8, Line 13-14, "This analysis shows how Europe has been steadily improving its agricultural GHG intensity (both in terms of unit of cropland and of unit of agricultural production value)". What is the meaning of improving emission intensity? Is there a problem with this

sentence? You said that Europe's emissions went significantly down, so I think the emission intensity was reduced.

**Answer:**

By 'improving' we meant 'decreasing'. But agree that it was not very clear and we edited it in the text. Thank you for highlighting it.

**Reviewer's comment:**

5. Although the authors have made a lot of improvements, some paragraphs are still chaotically divided. please check carefully.

**Answer:**

We reviewed the text one more time and found some minor opportunities to improve readability (such as by improving punctuation). Thank you for pointing it out. We trust the revised text is now acceptable.